# The Effects of Deoxynivalenol on the Ultrastructure of the *Sacculus Rotundus* and *Vermiform Appendix*, as Well as the Intestinal Microbiota of Weaned Rabbits

**DOI:** 10.3390/toxins12090569

**Published:** 2020-09-04

**Authors:** Chunyang Wang, Libo Huang, Pengwei Wang, Quancheng Liu, Jinquan Wang

**Affiliations:** 1Shandong Provincial Key Laboratory of Animal Biotechnology and Disease Control and Prevention, College of Animal Science and Veterinary Medicine, Shandong Agricultural University, 61 Daizong Street, Taian 271018, China; wcy@sdau.edu.cn (C.W.); huanglibo@sdau.edu.cn (L.H.); 2018110403@sdau.edu.cn (P.W.); 2019110441@sdau.edu.cn (Q.L.); 2Feed Research Institute, Chinese Academy of Agricultural Sciences, Beijing 100081, China

**Keywords:** deoxynivalenol, ultrastructure, *sacculus rotundus*, *vermiform appendix*, intestinal microflora, weaned rabbit

## Abstract

Deoxynivalenol (DON) is a mycotoxin found in grains that poses a potential threat to human and animal health, and the gastrointestinal tract is the primary target organ. There are few studies focused on the toxicology of DON to rabbits, especially on the relation among DON, microbiota, and the gut-associated lymphoid tissue. A total of 30 weaned rabbits (35 d) were evenly divided into the control group and DON group (1.5 mg/kg bodyweight (BW)) based on their body weight. After a 24-day trial, the ultrastructures of the *sacculus rotundus* and *vermiform appendix* were observed using a scanning electron microscope and transmission electron microscopy. The morphology and microflora in the ileum, caecum, and colon were also examined. The results proved that the ultrastructure of the *sacculus rotundus* and *vermiform appendix*, as well as the integrity of the intestinal barrier (especially for the ileum), were impaired after DON was administrated to the rabbits. Compared to the control group, the relative abundance and diversity of the microflora decreased in all three intestinal segments in the DON group, particularly in the ileum and caecum. In conclusion, the toxic effect of DON on weaned rabbits may be performed by destroying the structure of the *sacculus rotundus* and *vermiform appendix*, as well as affecting the structure and diversity of the intestinal flora.

## 1. Introduction

Deoxynivalenol (DON), mainly synthesized through toxin-producing fungus of the Fusarium genus, is found in grains all over the world, in particular, in wheat, barley, corn, and their by-products [1]. As one of the mycotoxins with the highest positive rate, DON has a variety of toxic effects on humans and animal species through the consumption of food and feed [2]. Commercial feedstuffs are commonly used in rations for rabbits, accounting for 60–70% of production costs [3]. The complete feeds of rabbits are formulated by different ingredients, including grains, and the contamination of mycotoxins (such as DON) has become an unavoidable problem with the continuous expansion of the rabbit breeding scale [4,5].

The intestine is one of the primary target organs attacked by mycotoxins following the intake of DON [6,7]. Research proved that DON can inhibit the absorption of intestinal nutrients, change the function of intestinal cells, and damage the intestinal barrier [8,9]. The gut-associated lymphoid tissue (GALT) is a mature constituent part of the mucosa immunity system, which can promote the maturation of the immune system after birth, as well as participate in the resistance to pathogens and protection for hosts [10]. Compared to other mammals, the structure of the GALT of rabbits is more developed, with two kinds of unique immune lymphoid organs, the *sacculus rotundus* (SR) and *vermiform appendix* (VA) [11,12], which are related to the intestine and considered to be multifunctional organs integrating immune, digestive, and secretory functions, accounting for more than 50% of the total lymphoid tissues for rabbits [13,14]. The functions of the SR and VA are closely related to the morphological structure [15,16]. At present, the studies concerning the immune toxicity induced by DON mainly focused on pigs, chickens, and certain cell lines, while the research on rabbits is very limited.

Intestinal microflora acts as a vital role in protecting host health, such as the energy intake in the diet, the production of key metabolites for the host, the development and maturation of the immunity system, and the response to intestinal diseases [17]. The caecum of rabbits is very large, containing nearly 40% of the intestinal contents, and the type and number of intestinal flora are relatively abundant, which can not only digest the high-fiber diet effectively, but also stimulate the innate immune system, which affects the overall health of rabbits [18,19]. Researchers reported that changes in the gut microbiome are not only a sign of disease, but also actively promote the pathogenesis of intestinal disease [20], and some studies also demonstrated that the gut microbiota of rabbits is closely related to the maturation of the humoral immunity system and GALT [21,22,23]. It is meaningful to investigate the relation among DON, microbiota, and the GALT of rabbits.

Considering the points mentioned above, the aim of the present study was to investigate the effect of DON on the ultrastructure of the SR and VA, the intestinal morphology, and intestinal microflora in weaned rabbits.

## 2. Results

### 2.1. Effects of DON on Morphology of the SR and VA via Scanning Electron Microscopy (SEM)

Representative morphologies of the SR by SEM are illustrated in Figure 1a. The photographs showed that the microvilli on the mucosal epithelial surface of the SR were complete, neatly arranged, dense, and with clear intercellular boundaries in the control group. The demarcation lines between the follicles were well defined, and the epithelium of dome epithelium (DE) was neatly arranged. From the longitudinal section, the microvilli of the SR were neatly arranged, and the boundary between the dome epithelium (DE) and the villi epithelium was clear without adhesion in the control group. After DON was administrated to weaned rabbits, obvious destruction in the mucous epithelium of the SR was observed, including mucosal epithelial cells that were exfoliated and lost extensively, many short and loose villi and microvilli appeared, and the microvilli were loosely arranged and no longer complete and tidy. Additionally, compared with the control group, the distribution of microfold cells (M cells) in the epithelium of the DON group did not increase significantly. The longitudinal photographs showed that the height of the villi was significantly shorter and irregular, accompanying the disorder of the sub-mucosal lymphoid tissue as well in the DON group.

Meanwhile, the SEM photographs of the VA (Figure 1b) showed that, after the administration of DON, the ultrastructure of the mucous epithelium of the VA was also obviously damaged, and it had the same tendency as the destruction of the SR. The distribution of M cells in the VA increased obviously compared with those in the SR after DON administration. The villus barriers in both of the SA and VA were damaged and exhibited histological lesions.

### 2.2. Effects of DON on Morphology of the SR and VA via Transmission Electron Microscopy (TEM)

Representative morphologies of the SR of rabbits by TEM are illustrated in Figure 2a. The photographs show that, in the control group, the mucosal epithelial cells were arranged neatly, the intercellular structure was complete and tightly connected, and the morphology of the immune cells was normal. However, after the consumption of DON, the ultrastructure of the VA was broken and damaged, the microvilli became shorter or even broken, the intercellular connections were not well formed or were arranged loosely, and part of the cell structure was deformed, particularly for lymphocytes.

Representative morphologies of the VA by TEM are illustrated in Figure 2b. For the control group, the typical M cells are observed on the surface of the lumen of the same lymphoid follicle, and these M cells are connected with neighboring cylindrical cells through desmosomes and apical tight junctions. After treating the rabbits with DON, the microvilli of the VA were broken and damaged, the intercellular connections were broken, and the cells are not arranged tightly and well-spaced. The lymphocytes and plasma cells were deformed, and many vacuoles and black particles were observed in the lymphocytes. In brief, DON damaged the ultrastructure of the SR and VA of weaned rabbits.

### 2.3. Effects of DON on the Intestinal Morphology of Weaning Rabbits

The typical morphological photographs of the ileum, caecum, and colon of rabbits via SEM are illustrated in Figure 3. The photographs showed that the intestinal epithelial surfaces in the control group were complete, neatly arranged, and dense. However, after DON was consumed by rabbits, the mucosal epithelium of three intestinal segments, in particular for the ileum, was exfoliated and lost extensively, with irregular microvilli. Among them, the mucosa of the colon had the least damage compared with that in the caecum and ileum.

### 2.4. Analysis of Operational Taxonomic Units (OTUs) and the Alpha Diversity of Samples

In this experiment with two groups, three rabbits were chosen randomly from each group, and three intestinal segments were sampled in each rabbit, and, therefore, the DNA of 18 samples was extracted and sequenced totally. Then, six group libraries were obtained after 18 raw data were analyzed, and they were coded as CI, CM, CJ, DON-I, DON-M, and DON-J, that is, C and DON representing the control group and DON group respectively, as well as I, M, and J representing the samples from the jejunum, duodenum, and caecum, respectively.

In the present study, an average of 89,265 effective sequence reads were acquired from each group library based on 16S rDNA amplicon sequencing, wherein the smallest number of reads was 81,336 and the largest was 92,293 (Table 1). Alpha diversity is typically applied to analyze the richness and diversity of the microbial community for one sample (within-community). In this paper, several statistical analysis indexes, including ACE, Shannon, Chao, Simpson, observed species, and PD-whole-tree, were applied to evaluate the diversity of microbial communities within each sample (Table 1). The results proved that, in the control group, the flora diversity in the caecum was the most abundant, followed by the colon and jejunum in the control group. After DON was added to the rabbits, the flora diversity in the ileum and particularly in the caecum, were clearly decreased. The number of observed species in the caecum and colon were all decreased in the DON group compared with the control group.

The rarefaction curve and rank–abundance curve were used to describe the richness and diversity of the samples in a group (Figure 4). Rarefaction curves were built according to the amount of sequence data extracted and the number of corresponding species. The tendency of the rarefaction curve was flattened out, indicating that the data volume of sequencing in this study was appropriate, that is, only a minority of new species (OTUs) were produced even if more data were detected.

The rank–abundance curve was drawn based on the relative abundance of OTUs as the ordinate, and the sequence numbers as the abscissa, which can intuitively reflect the abundance and uniformity of the species within the sample. The smoother the rank curve in the vertical direction, the more even distribution of species, while the larger the span of the rank curve in the horizontal direction, the higher the richness of species. The results of the rank–abundance curve showed that almost all samples were close to the saturation platform, which indicates that the present data have a sufficient depth to capture most of the diversity information of the samples.

### 2.5. Annotation of Intestinal Bacterial Flora of Weaned Rabbits

The relative abundance of the dominant bacteria floras in six libraries at the phylum and family levels are displayed in Figure 5. As shown in Figure 5a, *Firmicutes*, *Bacteroidetes*, and *Proteobacteria* were the main intestinal bacteria at the phylum level for all of the samples of rabbits. The relative abundance of *Proteobacteria*, *Actinobacteria,* and *Cyanobacteria* in both the ileum and caecum of rabbits decreased significantly after DON administration, while that of *Firmicutes* and *Bacteroidetes* in the ileum and colon increased significantly. However, when compared to the control group, there was no obvious variation for the abundance of bacteria floras in the colon in the DON group animals at the phylum level. In Figure 5b, the relative abundance of *Ruminococca* in the ileum and colon of rabbits was upregulated significantly after DON administration, while the abundance in the colon remained stable.

The top 30 bacteria floras with the highest relative abundance at the genus level are displayed in Figure 6. The relative abundance of *Ruminococcaceae*, *Bacteriods,* and *Lachnospiraleaes* all increased significantly in the ileum, caecum, and colon of rabbits after DON administration. *Ruminococcaceae* represented the largest number of bacteria in the three intestinal segments at the genus level. In addition, the relative abundance of the “others” significantly decreased in the ileum, caecum, and colon of DON group rabbits, indicating that the variety and richness of the intestinal microflora were reduced after the addition of DON to rabbits, especially in the ileum and caecum.

### 2.6. Beta Diversity

The diagram of the Unweighted Pair-Group Method with Arithmetic Mean (UPGMA) clustering tree based on the Unifrac weighted distance is illustrated in Figure 7. Each group was roughly clustered, indicating that the similarity of samples within the group was higher than that between groups. In addition, some samples had crossovers, indicating that there were also differences in the composition and structure of the microflora between individuals within the group as well as between two groups.

## 3. Discussion

### 3.1. DON and the Intestinal Barrier

The gastrointestinal tract is the primary target organ attacked by mycotoxin-contaminated food or feed [24]. After mostly being absorbed by the small intestine, DON can cross the physiological barrier of the intestine, enter the bloodstream, and thus affect the functions of histocytes [25,26]. DON can break down the intestinal barrier of multiple species and cause intestinal lesions, subsequently resulting in the reduction of production performance and the impairment of health [27,28,29]. The intestinal wall of rabbits is thin, and the environment of the intestine is relatively fragile and susceptible to changes in feed and the invasion of harmful bacteria to cause intestinal diseases, especially inflammatory intestinal diseases [30].

In the present study, SEM technology was used to observe the integrity of the intestinal barrier of weaned rabbits, and we found that the mucosal epithelium was exfoliated and lost extensively, with irregular microvilli. The mucosa in the colon had the least destruction compared with in the caecum and ileum, indicating that DON caused intestinal damage in rabbits through destroying the integrity of the physiological barrier in the whole intestines but with different degrees of severity. SEM technology also was used to observe the intestinal morphology of piglets in a previous study, and the results illustrated that the histological structure of the mucosa in different intestinal segments was destroyed after the addition of DON to pigs [31].

Unlike our previous study in pigs, the destruction in the intestinal barrier of the rabbits induced by DON in this experiment was milder and was mainly in the ileum, while the damages to the caecum and colon were less. This may be because the rabbits are a typical post-fermented animal, and their intestinal flora are very rich, particularly in the caecum and colon, which could reduce the toxicity of DON by converting DON to de-epoxidation deoxynivalenol (DOM) [32]. Based on the results of 16S RNA sequencing, the abundance and richness of the intestinal flora in the ileum of the rabbits were changed compared to the three intestinal segments, while the colon was relatively stable. These results suggested that DON can affect the intestinal flora and may further affect the absorption of DON in the intestinal tract, which is closely related to the toxic effect of DON.

### 3.2. DON and GALT of Rabbits

The SR and VA, as the unique GALT organs of rabbits, are primarily responsible for the development and maturation of B lymphocytes, as well as the transformation of secondary lymphoid tissues for adult rabbits, such as peyer patch (PPs) [33,34]. Studies proved that SR and VA can prevent and resist antigens from invading the body and deep lymphoid tissues, as well as protect the intestinal health [15,16]. The function of the SR and VA are closely related to its morphological structure; however, there are very few studies focused on the effect of DON on the ultrastructure of the SR and VA. In this study, we conducted both SEM and TEM to observe the morphology of SR and VA, and we found that the ultrastructures of the SR and VA were destroyed after DON was administrated to weaned rabbits as the microvilli became shorter or even broken, the intercellular connections were not well formed or were arranged loosely, part of the cell structure was deformed, and so on.

M cells usually enfold lymphocytes and macrophages and play a key role in transporting antigens from the enteric cavity to mucosa-associated lymphoid tissues, such as SR and VA, and certain pathogens can use M cells to invade the host and initiate infections [35,36]. The results from SEM and TEM in the present study illustrated that the distribution of M cells in the VA of weaned rabbits was more prominent than in the SA in control groups. However, after DON administration to rabbits, the distribution of M cells in the VA increased significantly, with no clear change in the SR. This may be because the antigen transportation mediated by M cells was more important in the VA than in the SA for rabbits, and thus the responses to DON in the VA were more sensitive than those in the SA. In consideration that DON can modulate the immune response of GALT, we suggested that the inflammatory reaction induced by DON may relate to the interaction between epithelial cells and intestinal immune cells [7,8,37,38]. In brief, we considered that DON may affect the immune function of rabbits and induce intestine disease by destroying the structure of the SR and VA.

### 3.3. DON, Intestinal Microbiota, and GALT

Rabbits are hindgut-fermentation herbivores with hundreds of microorganisms inhabiting the intestines, which constitute the intestinal micro-ecosystem and maintain a dynamic balance under the intervention of the intestinal immune system [39]. The stability and richness of the intestinal microbial community are key to maintaining the intestinal health of rabbits [40]. Increasing numbers of studies have proved that DON may alter the diversity and richness of the intestinal microflora, and the variation of the microbiome not only represents the occurrence of disease, but also promotes the development of disease [8,9].

In this study, the results of 16s RNA sequencing proved that the relative abundance of *Proteobacteria*, *Actinobacteria,* and *Cyanobacteria* in both the ileum and caecum of rabbits decreased significantly after DON administration, while that of *Firmicutes* and *Bacteroidetes* in the ileum and colon increased significantly. However, compared to the control group, there was no significant change for the abundance of bacteria flora in the colon in the DON group animals at the phylum level. The abundance of *Ruminococca* in the ileum and colon of rabbits was significantly upregulated at the family levels after DON administration, while those levels in the colon remained stable.

In addition, the relative abundance of *Ruminococcaceae*, *Bacteriods,* and *Lachnospiraleaes* all increased significantly at the genus level, and the relative abundance of the “others” significantly decreased in the ileum, caecum, and colon of the DON group rabbits. In all, the abundance and richness of the intestinal microflora in the ileum and caecum were reduced compared with the three intestinal segments after the addition of DON to the rabbits, while those in the colon were kept relatively stable. These data correspond to the morphological results of the intestinal barrier observed by SEM, that is, DON had the greatest impact on the ultrastructure of the ileum of weaned rabbits, and had the least damage to the colon. DON can affect the intestinal flora of weaned rabbits, and may further affect the absorption of DON in the intestinal tract as well as the integrity of the intestinal barrier.

The intestinal flora not only participates in the processing of nutrients and regulates the formation of intestinal blood vessels, but also promotes the development of the GALT and mucosal immunity, induces immune tolerance, as well as produces a diverse spectrum of pre-immune antibodies [41]. The intestinal microbiome can act as a signal hub linking the immune system and intestinal microflora, and their abnormal communication may give rise to complex diseases [42,43]. We proved that DON could lead to the variation in the abundance and diversity of intestinal flora in rabbits, and the destruction in the ultrastructure of the SR and VA, which act as specific GALT of rabbits. The intestinal microbiota exists in a reciprocal balance with the GALT, and the two systems affect one another to protect the rabbit’s health. When DON is taken up by rabbits, the balance is destroyed, which will harm the health of the rabbits. However, it is still unclear how the gut microbiota establishes a connection with the GALT, and the mechanism of these reactions also requires the further research.

## 4. Conclusions

In conclusion, the toxic effect of DON on weaned rabbits may occur via destruction of the structure of the *sacculus rotundus* and *vermiform appendix*, as well as by affecting the structure and diversity of the intestinal flora.

## 5. Materials and Methods

### 5.1. Ethics Statement

The protocol concerning animal experiments was audited and approved by the Animal protection committee of Shandong Agricultural University (ACSA-2018-032-September).

### 5.2. Animals and Sampling

The complete feed of weaned rabbits was provided by the Liuhe Feed Company (Shandong, China), and the ingredient composition, nutrition level, and the concentration of Aflatoxin B1 (AFB1), Zearalenone (ZEA), and DON, are listed in Table 2.

Thirty healthy, weaned Rex Rabbits (35-day) were divided evenly into the control group and DON group based on their initial weight, to provide 15 rabbits in each group. All rabbits in the two groups were fed with basic diets, but those in the DON group had DON standard (Triplebond Co., Guelph, ON, Canada) added into the drinking water. The additional amount of DON was 1.5 mg/kg BW/d, and detailed addition steps were performed according to the method reported by Yang et al. [29]. All experimental rabbits were fostered individually in standard metabolic cages for rabbits, which were placed in a professional rabbit nutrition research laboratory with the room temperature maintained between 18 and 28 °C. This animal trial was conducted for a total of 31 days, of which the pre-feeding period was 7 days and the formal trial period was 21 days. On day 24, five rabbits of each group were selected stochastically and euthanized by injection, then the samples of the *sacculus rotundus*, *vermiform appendix*, and intestinal tract (including the ileum, caecum, and colon) were collected quickly, rinsed using 0.9% saline, and then fixed by 2.5% glutaraldehyde for later electron microscope examination. The content of the homologous three intestinal segments were collected and immediately stored at −70 °C until the 16S DNA sequencing analysis

### 5.3. Scanning Electron Microscope

After approximately two-centimeter segments of the SR, VA, and intestinal tract were collected respectively and fixed in 2.5% glutaraldehyde-paraformaldehyde overnight at 4 °C, the samples were washed using 0.2M phosphate-buffered saline (PBS) three times, 30 min per time, and then passed through a series of graded alcohols for dehydration, i.e., 30%, 50%, 70%, 80%, 90%, 95%, and 100% ethanol. After being fixed in isoamyl acetate for 24 h, the samples were dried near the critical point of CO_2_, and then adhered to the sample stage with a conductive adhesive under the stereo microscope. Finally, the samples were observed and photographed under a scanning electron microscope (HITACHI S-570, Tokyo, Japan) after metal spraying with an Ion Sputter Coater (SBC-12, Beijing, China).

### 5.4. Transmission Electron Microscopy

After about one-cubic-millimeter pieces of the SR and VA were cut and immersed in 2.5% glutaraldehyde-paraformaldehyde over 24 h, the samples were fixed in 1% osmium acid for 1 h, and then washed thoroughly with PBS. Ultra-thin slices were prepared according to the conventional technique and dyed using double staining with uranium acetate and lead citrate. The slices were observed and photographed using a transmission electron microscope (JEM-1400, Tokyo, Japan).

### 5.5. DNA Extraction and 16S rDNA Amplicon Sequencing

We weighed 200 mg intestinal content samples and added them into 1.4 mL of Argininosuccinate Lysis buffer (Qiagen, Hilden, Germany). Then, the QIAamp DNA Stool Mini kit for feces samples (Qiagen, Hilden, Germany) was used to extract the DNA following the instructions. The DU640 Nucleic Acids and Protein Analyzer (Beckman Coulter, Brea, CA, USA) was applied to evaluate the quality of the DNA at a wavelength of 260/280. Finally, after germfree ultrapure water was used to dilute and quantify precisely to 1 ng/uL, all the DNA samples were stored at −20 °C for later use.

16S rDNA Amplicon Sequencing was applied to sequence a polymerase chain reaction (PCR) product or captured fragment of a specific length. First, the barcoded primer pair 341F/806R set was designed for PCR amplification in this study, and the 341F primer was the following: CCTAYGGGRBGCASCAG, while the 806 primer was the following: GGACTACNNGGGTATCTAAT. The diluted genomic DNA was used as template DNA, and the barcoded primers were applied to amply the V3–V4 fragments of the 16S rDNA gene. The detailed steps of PCR reaction were performed on the basis of the method reported by Caporaso et al. [44]. All the procedures of 16S rDNA Amplicon Sequencing were performed based on the IonS5^TM^XL sequencing platform (Novogene, Beijing, China).

### 5.6. Bioinformatics Analysis

A small fragment library for single-end sequencing was constructed based on the IonS5^TM^XL sequencing platform, and clean reads with high quality were obtained through the cutting and filtering of the initial reads. Uparse software (version v7.0.1001, Edgar, Tiburon, CA, USA) was used to cluster the clean reads of all samples, and the sequences with 97% identity by default were considered as OTUs. During the data analysis, three replicate samples were aggregated into a total database. Therefore, 18 samples were aggregated into six databases, which were coded as CI, CM, CJ, DONI, DONM, and DONJ.

The Mothur method and SSUr RNA database were used to representatively annotate species for the OTU sequence (set threshold 0.8–1), and the taxonomic information of each sample was generated into the phylum, class, order, family, and genera levels. To explore the differences of community structure between different samples or groups, multiple sequence alignment was performed based on the OTUs, and a phylogenetic tree was constructed through dimensionality reduction analysis, such as principal co-ordinates analysis (PCoA) and principal component analysis (PCA). Finally, all data were applied in the analysis of both the Alpha diversity and Beta diversity, which must be normalized in advance.

Several Alpha diversity indexes, including ACE, Simpson, Shannon, Chao, and Observed species, were calculated using QIIME software, while the rarefaction curves and rank–abundance curves were drawn and analyzed using R software. The graphs of PCA, PCoA, and NMDS were also drawn and analyzed using R software (version 2.15.3, R Core Team, Vienna, Austria, 2016). In this study, our analyses of the differences in the Alpha diversity indexes and Beta diversity indexes between groups were performed using the parametric test and non-parametric test, respectively. The T-test and Wilcox test were applied to analyze the differences between the two groups, while the Tukey test and Wilcox test were used to analyze among more than three samples [45].

## Figures and Tables

**Figure 1 toxins-12-00569-f001:**
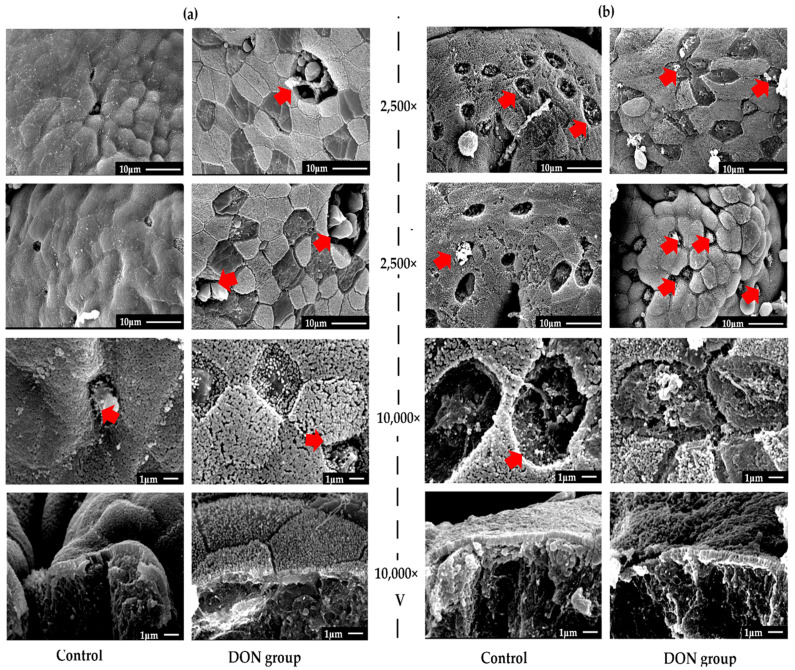
The effect of deoxynivalenol (DON) on the morphology of the *sacculus rotundus* (**a**) and *vermiform appendix* (**b**) via scanning electron microscopy (SEM) (*n* = 3). 2500× and 10,000× represent the magnification of electron microscopy, and V represents the photo of a vertical section magnified. The microfold cells are marked by red arrows.

**Figure 2 toxins-12-00569-f002:**
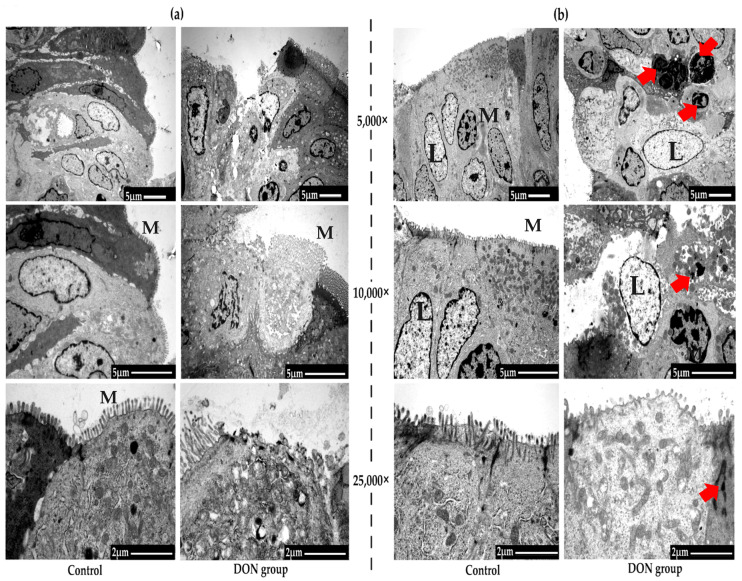
The morphology of the *sacculus rotundus* (**a**) and *vermiform appendix* (**b**) in two groups via transmission electron microscopy (TEM) (*n* = 3). 5000×, 10,000×, and 25,000× represent the magnification of electron microscopy. M means microfold cell and L means lymphocytes. Typical lesions of cells are marked by arrows.

**Figure 3 toxins-12-00569-f003:**
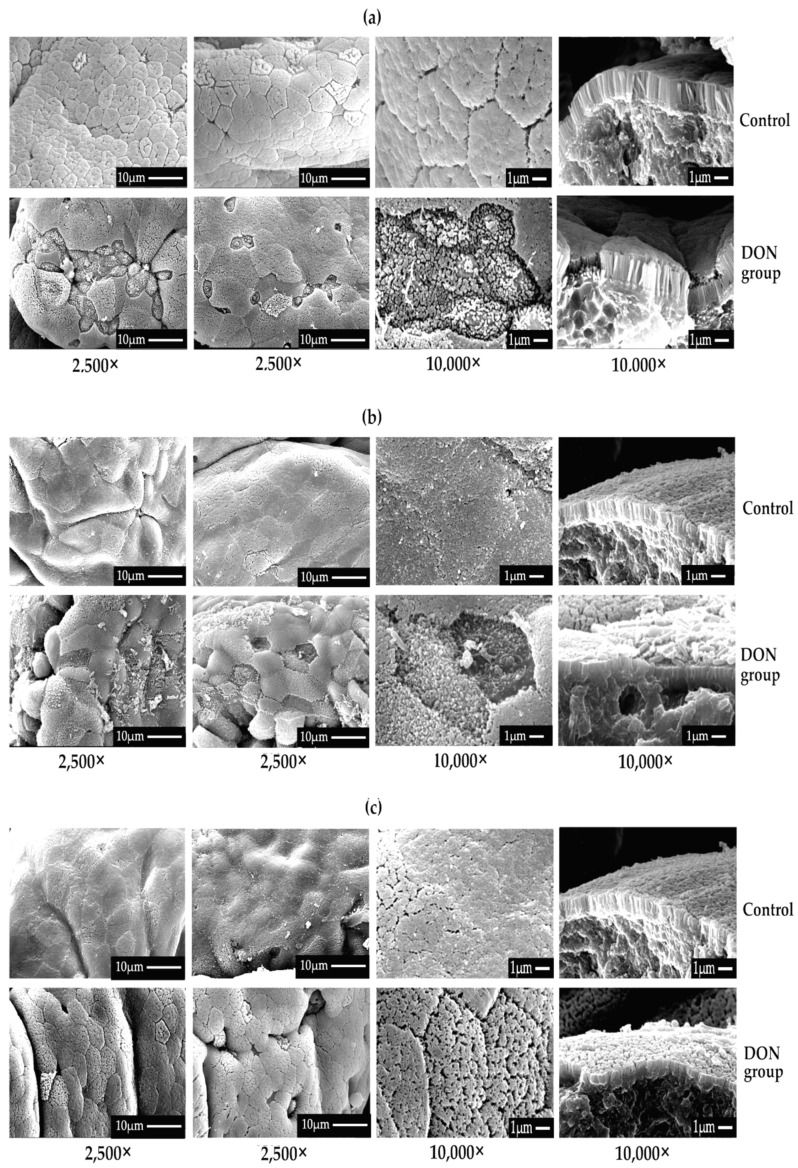
The effect of DON on the intestinal morphology of weaned rabbits via TEM (*n* = 3). (**a**–**c**) represent photographs of the ileum, caecum, and colon respectively, in two groups. 2500× and 10,000× means the magnification of electron microscopy. The pictures in column 4 represent the photo of a vertical section magnified.

**Figure 4 toxins-12-00569-f004:**
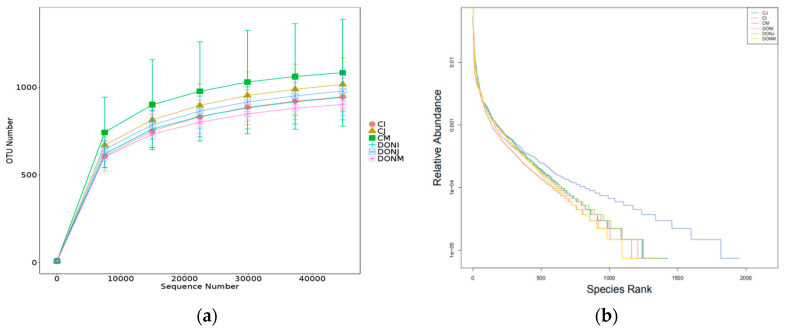
Rarefaction curve (**a**) and rank–abundance (**b**) curve in six group libraries. For the rarefaction curve, the *x*-axis refers to the number sequencing reads that were randomly chosen from a certain sample to obtain OTUs, and the *y*-axis refers to corresponding OTUs. For the rank–abundance curve, the *x*-axis is the abundance rank, and the *y*-axis is the relative abundance.

**Figure 5 toxins-12-00569-f005:**
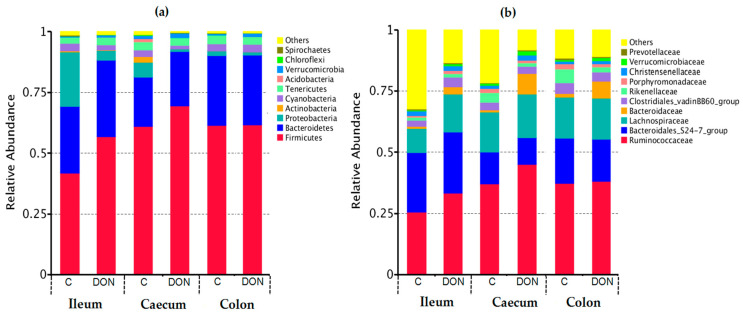
Relative abundance of the dominant species in six libraries at the phylum (**a**) and family (**b**) level (top 10). Each bar represents the relative abundance of each sample. Each color represents a particular bacterial family. Sequences that could not be classified into the top 10 were classified as ‘others’. C and DON represent the samples from the control group and DON group.

**Figure 6 toxins-12-00569-f006:**
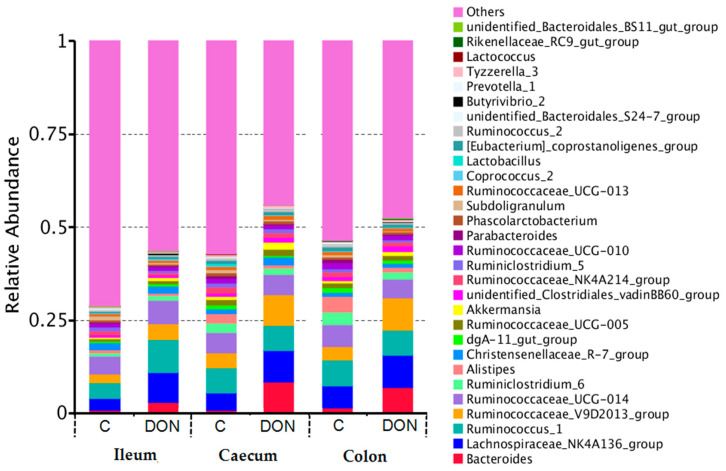
The relative abundance of the dominant bacteria at the ileum, caecum, and colon of rabbits at the genus level (top 30). Each bar represents the relative abundance of each sample. Each color represents a particular bacterial family. Sequences that could not be classified into the top 30 were classified as ‘others’. C and DON represent the samples from the control group and DON group.

**Figure 7 toxins-12-00569-f007:**
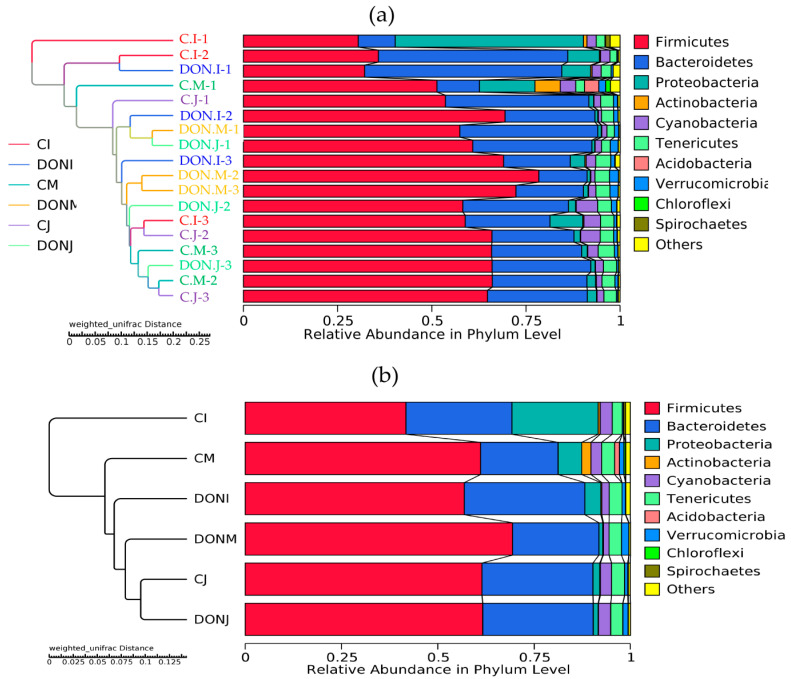
The Unweighted Pair-Group Method with Arithmetic Mean (UPGMA) cluster tree using a un-weighted unifrac analysis at the phylum level. (**a**) The data from 18 samples via 16S rRNA gene sequencing and (**b**) the data from six group libraries. C and DON represent the samples from the control group and DON group, respectively. I, M, and J represent the samples from the ileum, caecum, and colon, respectively.

**Table 1 toxins-12-00569-t001:** Operational Taxonomic Unit (OTU) ^1^ data and the alpha diversity of six group libraries from 16S rDNA amplicon sequencing.

Sample Name	Raw PE	Qualified	AvgLen (nt)	Effective (%)	ACE	Shannon Ave	Chao Ave	Simpson	Observed Species	PD-Whole-Tree
C-I	81,336	72,142	372	85.00	1022	6.03	1003	0.86	945	62.5
DON-I	92,115	82,606	372	86.77	1025	7.01	1023	0.97	943	61.5
C-M	90,527	80,535	372	84.03	1141	7.74	1125	0.99	1083	71.7
DON-M	91,563	81,786	372	84.26	972	6.94	955	0.96	902	57.8
C-J	89,780	79,680	372	83.59	1110	7.52	1097	0.98	1016	65.5
DON-J	87,995	78,616	372	84.55	1061	7.14	1045	0.97	978	65.5

^1^ OTUs was classified at the 3% dissimilarity level. Raw paired-end reads (PE) refers to original PE reads detected via IonS5^TM^XL sequencing platform. Qualified refers to the clean reads with high-quality sequences. AvgLen means the average length of valid tags. Effective refers to Effective Tags/Raw PE × 100%. The observed species, as well as the indexes of community richness (Chao, ACE), community diversity (Shannon and Simpson), and phylogenetic diversity (PD_whole_tree) were calculated using the QIIME software. CI, CM, CJ, DON-I, DON-M, and DON-J, with C and DON representing the control group and DON group respectively, as well as I, M, and J representing the samples from the jejunum, duodenum, and caecum, respectively.

**Table 2 toxins-12-00569-t002:** Component, nutrition level, and mycotoxin content ^1^ of the basal diet.

Component (%)		Nutrition Level (Calculated)	
Corn-germ powder	19	Digestible energy (MJ/kg)	10.06
Bean pulp	17	Total Phosphorus	0.55
Corn	14	Calcium	0.72
Wheatbran	13	Crude fat	3.34
Rice husk	10	Crude ash	10.45
Alfalfa	10	Crude fiber	18.78
Soybean straw powder	7	Crude protein	20.51
Malt Sprout	5	Dry matter	88.64
Sweet wormwood	3.5		
Premix compound ^2^	1.5		
Total	100		
**Mycotoxin Content (μg/kg)**			
Aflatoxin B1 (AFB1)	7.08		
Zearalenone (ZEA)	257.76		
Deoxynivalenol (DON)	23.18		

^1^ Detected via high performance liquid chromatography (HPLC) methods. ^2^ Premix compound supplied diet per kg: Vitamin D 3100 IU; Vitamin A 12,000 IU; Vitamin B 63.52 mg; Biotin 0.2 mg; Choline 0.6 g; VE 50 mg; Fe 60 mg; Zn 60 mg; Cu 40 mg; Mn 9 mg; Se 0.2 mg; NaCl 5000 mg; Lysine 1500 mg; and Methionine 1000 mg.

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
