# Peer review of "The Effects of Deoxynivalenol on the Ultrastructure of the Sacculus Rotundus and Vermiform Appendix, as Well as the Intestinal Microbiota of Weaned Rabbits"

_toxins, 2020, doi:10.3390/toxins12090569_

Round 1

Reviewer 1 Report

The authors investigated the effect of deoxynivalenol on the ultrastructure of  Sacculus Rotundus and Vermiform appendixin, as well as the intestinal microbiota of weaned rabbit.

Line 5: Deoxyhumidol? I suppose this is a typo and you were meaning to write deoxynivalenol. 

Line 9: 1.5mg/kg.BW I think that fullstop between kg and BW is not necessary. 

Lines 21-22: the sentence "...rabbits may carried out by destroying..." is not clear. Please rephrase. You are repeating the same thing in the conclusion, too Please have an language expert look at your text. 

General remarks:

  1. sacculus rotundus, vermiform appendix Should not the latin expresions be in italic? 
  2. is DON or (its masked forms) measured in rabbits' blood?

In my opinion, this extensive research, even though rich in information and data, has a major flaw. ANd that would be the absence of DON and DON masked forms concentration in the sampled rabbits. You cannot be sure that the damage was done by DON, it can also be related to the presence of its masked forms. Sure, you gave DON to rabbits, but I think that this theory could have been proofed with simple blood test for DON and its derivates. 

Author Response

Dear Journal Manager:

Great thanks for your review. At first, the revised manuscript has been corrected by the professional english editor. Then revised portion are marked using red color in the paper. The main corrections and explain in the paper and the responds to the reviewer’s comments are as following:

Line 5: Deoxyhumidol? I suppose this is a typo and you were meaning to write deoxynivalenol. 

--- Great thanks for you find my fault. The word “Deoxyhumidol” has been changed to “Deoxynivalenol” in the revised manuscript.

Line 9: 1.5mg/kg.BW I think that fullstop between kg and BW is not necessary. 

--- Great thanks for you find my fault. The word “1.5mg/kg.BW” has been changed to “1.5 mg/kg bodyweight (BW) ” in the revised manuscript.

Lines 21-22: the sentence "...rabbits may carried out by destroying..." is not clear. Please rephrase. You are repeating the same thing in the conclusion, too Please have an language expert look at your text. 

--- Great thanks for your suggestion. This sentence has been modified in the revised manuscript. And the full text has been undergo english edit professionally in the revised manuscript.

General remarks:

sacculus rotundus, vermiform appendix Should not the latin expresions be in italic? 

--- Great thanks for your suggestion. sacculus rotundus and vermiform appendix have been writed in italic in the revised manuscript.

is DON or (its masked forms) measured in rabbits' blood?

In my opinion, this extensive research, even though rich in information and data, has a major flaw. ANd that would be the absence of DON and DON masked forms concentration in the sampled rabbits. You cannot be sure that the damage was done by DON, it can also be related to the presence of its masked forms. Sure, you gave DON to rabbits, but I think that this theory could have been proofed with simple blood test for DON and its derivates. 

--- Great thanks for your professional suggestion. The concentration of DON and DOM-1 in rabbits' blood, urine and feces were measured in our research, and part of data has been publised by Yang et al, 2019. (topic: The Effect of low and high dose deoxynivalenol on intestinal morphology, distribution and expression of inflammatory cytokines of weaning rabbit )

Sincerely yours,

Chunyang Wang

Reviewer 2 Report

Good job

please find my minor comments in the pdf file. they are minor but they have to be seriously considered in the revised version.

Good luck

Author Response

Dear Journal Manager:

Great thanks for your review. Thanks very for your comments, the manuscript has been revised, and corrected by professional english editor in the revised version. 

If there is any points need to be clarified, please tell me by Email.

Great thanks for your special work for this manuscript.

Sincerely yours,

Chunyang Wang